# Holistic acceptability of an adult levofloxacin formulation in children and adolescents on a tuberculosis preventive treatment trial

Susan E. Purchase[1]*, Dillon T. Wademan[1], Nosibusiso L. Tshetu[1], Mohhadiah Rafique[1], Graeme Hoddinott[1], James A. Seddon[1,2], H. Simon Schaaf[1], Anneke C. Hesseling[1]

**1** Desmond Tutu TB Centre, Department of Paediatrics and Child Health, Faculty of Medicine and Health Sciences, Stellenbosch University, Tygerberg, South Africa, **2** Department of Infectious Disease, Imperial College London, London, United Kingdom

* purchase@sun.ac.za

**Data Availability Statement:** The datasets generated and analysed during the current study are not publicly available due to the need to protect

## Abstract

Drug-resistant tuberculosis (TB) is threatening global TB control. Although formulations designed for children are a priority, adult levofloxacin formulations are widely used in TB treatment and prevention. TB-CHAMP was a cluster-randomised, placebo-controlled trial evaluating the efficacy and safety of 24 weeks of daily levofloxacin to prevent TB in child and adolescent household contacts of adults with infectious multidrug-resistant TB. Nested in-depth longitudinal qualitative work was conducted in a subset of children and their caregivers to understand broader experiences of treatment acceptability. We conducted 41 interviews with 8 caregivers of children <6 years, and with 6 older children responding for themselves. Children who could not swallow the adult formulation whole, found the tablet unpalatable, although they learnt to tolerate the taste over time. Most caregivers and children came from families with substantial experience of TB, but felt they knew little about TB preventive therapy. Many families experienced challenging socio-economic circumstances. Poor acceptability was mitigated by sympathetic study personnel, assistance with transport and financial compensation. The adult formulation of levofloxacin was disliked by many younger children but was acceptable to children able to swallow the tablet whole. In addition to using acceptable drug formulations, TB preventive treatment implementation models should include patient education and should accommodate patients' socioeconomic challenges.

## Introduction

Drug-resistant tuberculosis (TB) remains a global public health threat. Multidrug-resistant (MDR)-TB is defined as disease caused by strains of *Mycobacterium tuberculosis* with resistance to at least isoniazid and rifampicin. The World Health Organization (WHO) estimates that 410 000 people develop rifampicin-resistant/MDR-TB each year [1]. Modelling data suggest that 2 million children (<15 years old) globally are infected with MDR-TB, with 25,000–

participant confidentiality but are available on reasonable request. Requests can be directed to the Health Research Ethics Committee at Stellenbosch University (ethics@sun.ac.za).

**Funding:** TB-CHAMP was made possible thanks to Unitaid's funding of the BENEFIT Kids project to Stellenbosch University. Unitaid accelerates access to innovative health products and lays the foundations for their scale-up by countries and partners. Data collection for this study was funded by the South African National Research Foundation through a South African Research Chairs Initiative (SA NRF – Hesseling, SARChI). GH received the financial assistance of the European Union (Grant no. DCI-PANAF/2020/420-028), through the African Research Initiative for Scientific Excellence (ARISE), pilot programme. ARISE is implemented by the African Academy of Sciences with support from the European Commission and the African Union Commission. The contents of this document are the sole responsibility of the author(s) and can under no circumstances be regarded as reflecting the position of the European Union, the African Academy of Sciences, and the African Union Commission. The funders had no role in study design, data collection and analysis, decision to publish, or preparation of the manuscript.

**Competing interests:** The authors have declared that no competing interests exist.

32,000 children progressing to MDR-TB disease annually [2, 3]. Although children with MDR-TB have good treatment outcomes, drug therapy is long and challenging, involving multiple drugs with numerous potential adverse effects. Prevention of MDR-TB is therefore critically important.

TB preventive treatment (TPT) is universally recommended for young children exposed to drug-susceptible TB [4], with multiple WHO-recommended regimens available and several child-friendly formulations developed [5]. Until very recently, however, there was only a conditional recommendation from WHO to use fluoroquinolone-based TPT for contacts of MDR-TB, given the low-quality evidence available and lack of data from randomized trials [6]. As a result, implementation of TPT for the prevention of MDR-TB has been extremely limited. TB-CHAMP (ISRCTN92634082) was a community-based cluster-randomized phase 3 double-blind placebo-controlled trial evaluating the efficacy and safety of 24 weeks of daily levofloxacin to prevent TB in at-risk child and adolescent household contacts of adults with infectious MDR-TB [7]. The trial, implemented at five sites across South Africa, used a Macleods Pharmaceuticals 250mg, film-coated, scored, adult levofloxacin formulation with matched placebo. Participants were dosed at 15–20 mg/kg daily using standard WHO-recommended dosing weight bands. The novel 100mg dispersible formulation was not used in the trial due to results from a recent pharmacokinetic trial [8] indicating much lower bioavailability of the adult formulation compared with the dispersible formulation, and subsequent concerns regarding correct dosing in young children.

Although formulations designed especially for children are a priority, adult formulations are often adapted for use in children in TB care. Uptake and demand for new child-friendly formulations, even when developed, remains low due to high costs, lack of adequate prioritization of children and reluctance of some healthcare workers to adopt novel formulations [9]. Early quantitative data from TB-CHAMP (unpublished) is showing relatively good acceptability of the adult levofloxacin 250mg formulation, especially in children able to swallow the tablets whole. This formulation is already used extensively as part of TB treatment in children globally. We aimed to understand children's and caregivers' experiences of the 250mg levofloxacin formulation and the matched placebo.

## Methods

This nested qualitative evaluation was implemented in the TB-CHAMP trial and was conducted in a subset of child participants and their caregivers at a single site (Desmond Tutu TB Centre, Stellenbosch University).

### TB-CHAMP setting and population

TB-CHAMP was conducted at five sites in South Africa, between 2017 and 2023. At the Cape Town site, participants were recruited from >100 primary healthcare clinics and 9 hospitals across metropolitan Cape Town, Western Cape Province. Initially only children under 5 years were eligible for the trial; a protocol amendment (Version 3.0) was implemented in September 2021 and allowed for enrolment of children aged 5-17-years-old.

### Treatment

Study medications were manufactured by Macleods Pharmaceuticals (India) as levofloxacin or taste-matched levofloxacin-placebo 250mg scored tablets and were given daily for 24 weeks. At the initial visit, caregivers were shown the medication and given options (whole, halved, crushed, or softened in liquid) regarding how to administer it to the children in their care. They were asked to dose the children at roughly the same time each day, with no food

restrictions. They were also asked to sign a treatment adherence card each day as they administered the study medication.

## Sampling strategy

The sampling unit (participant group) was the child, their caregiver and any other family members who self-identified as part of their 'household'. We used maximum variation purposive sampling to maximize diversity in experience [10], and to ensure a representative sample of females and males, and age and language groups. As the TB-CHAMP trial was blinded, we were unable to sample by study arm. Unblinding occurred once the main trial was completed. We sampled to *a priori* thematic saturation (the point at which no new information relating to the conceptual framework themes covered was obtained from data collection, or new participant groups) [11]. This point was determined by the socio-behavioural researchers conducting the interviews in consultation with the senior researchers who were concurrently reading case descriptions, and was based on the first interview, primarily for logistic reasons.

## Data collection process

We used a case study, longitudinal design, comprising multiple interactions with each participant group over the course of 6 months. Trained socio-behavioural researchers (MR and NT) worked alongside the clinical study team to recruit participant groups (between 01 August 2022 and 08 December 2022) before their first clinical study visit. Participant groups were selected based on their age, gender and language demographic, and availability and willingness to consent. Baseline interviews took place at/soon after enrolment, after the first dose of study drug had been administered. To allow for understanding of the changes in experiences of acceptability over time, caregivers/child participants were interviewed at 3 serial time points. Subsequent interviews were conducted at times suitable to the study team and participants; interviews were usually at least a month apart. Interviews with caregivers and older child participants were facilitated using a semi-structured discussion guide (S1 Text) conducted in the participants' preferred language (English, Afrikaans, or isiXhosa). The discussion guide was used flexibly around five topic areas, namely, 1) household and social context, 2) experience of TB disease and preventive treatment, 3) stigma and health beliefs, 4) perceptions of TB's impact on the future, and 5) reflections on trial participation. Interviews lasted 30–60 minutes and included verbal and activity-based probes adapted from participatory research methodology, expressly used to facilitate children's active participation in the study [12–14]. All discussions were audio recorded and researchers kept detailed observational and reflexive notes and copies of written/drawn activities completed. MR and NT wrote detailed case descriptions after each interaction, which were iteratively refined through multiple reviews by senior researchers (SP and DW) to inform future interactions with participants [15].

## Data analysis

Data analysis was iterative and followed a deductive two-step thematic approach. From the written case descriptions, within and across case analysis was used to identify unique and shared experiences across participants [16]. Where relevant, excerpts from recordings were transcribed and translated to illustrate key points. Analytic themes were informed by Wademan et al.'s (2022) conceptual framework of TB treatment acceptability [17].

### Ethics and informed consent

This sub-study was approved by the Health Research Ethics Committee, Stellenbosch University (N22/04/046 Sub-study M16/02/009). Written informed consent was provided by parents/legal guardians with additional assent in older children (7-17-years-old). All identifying data were removed to ensure confidentiality.

## Results

We conducted a total of 41 interviews with 8 caregivers reporting on children <6 years and 6 older children responding for themselves (Table 1). Of the 8 caregivers, 4 had children on placebo, and of the 6 older children, 2 were on placebo. All caregivers were female, reflecting current childcare patterns in South Africa [18]. None of the caregivers were index cases.

Data analysis was guided by the three domains of Wademan et al.'s acceptability framework [17] and is summarised in Fig 1.

### Usability: Palatability, administration, appeal

Children and caregivers described both the levofloxacin and the placebo as bitter (like aloe/lemon/paracetamol/dark chocolate), and generally disliked the taste. An 8-year-old summed up many participants' sentiments saying: "*They [the pills] are horrible!*". A caregiver described how her 2-year-old "*always runs away when she sees me take it [study medication] out*". One caregiver (mother to a 2-year-old) spontaneously identified the taste of the drug as a barrier to administering/taking the treatment. Older children noted that the tablets only tasted bad when they sucked them or left them on their tongue and said there was no taste when the tablets were swallowed whole. These older children found taking the drug easy, although two participants (aged 8 and 17) did comment that the drug smelled bad. One 3-year-old on levofloxacin said "*It's lekker (nice)*" but this child swallowed his tablets whole.

Caregivers of younger children used a variety of methods to ease administration. Many either crushed the tablet or softened it in water on a spoon, though a caregiver to a 3-year-old noted "*it takes about 20 minutes to dissolve*". One caregiver needed to divide the tablets and found breaking them in half the most difficult aspect of administration–however a 12-year-old noted they were easy to divide with scissors. Some caregivers mixed the tablets with breast milk, sweet tea, porridge or yoghurt, but generally found this did not mask the taste. Four caregivers needed to bribe their young children using sweets or money. Some caregivers commented on how the children learnt to tolerate the taste and size of the tablets over time. Several children watched other family members in the home swallow TB treatment and one caregiver noted that having his own tablets to swallow made her child (7-years-old) "*feel grown up*" and made the tablets more appealing.

### Receptivity: Adverse consequences, conceptions of health and illness, prior experiences of treatment and care

Most caregivers and children interviewed came from families with substantial experience of TB. In one family there were eight extended family members who had been diagnosed with

**Table 1. Description of participants stratified by caregiver or child/adolescent, by gender and by home language (language in which interview took place).**

| | Caregivers of children 0–6 years | | Children/adolescents 7–17 years | | |
|---|---|---|---|---|---|
| | Afrikaans | isiXhosa | Afrikaans | isiXhosa | Total |
| **Female** | 5 | 3 | 2 | 2 | 12 |
| **Male** | 0 | 0 | 1 | 1 | 2 |
| **Total** | 5 | 3 | 3 | 3 | 14 |

| USABILITY | RECEPTIVITY | INTEGRATION |
|---|---|---|
| **Palatability**<br>When crushed or softened described as: "*Bitter*" like "*aloe*"; "*lemon*", "*paracetamol*"; "*dark chocolate*"<br>When swallowed whole described as: "*Lekker*" (nice); "*no taste*"; "*Taste bad if you suck them*"<br><br>**Administration**<br>Process challenges:<br>Crushed/softened in water for young children<br>"*Takes about 20 minutes to dissolve*"<br>Halving tablet is difficult<br>Difficult to mask taste<br>Needed to bribe<br>Learnt to tolerate over time<br><br>**Appeal**<br>"*Feels grown up*" when taking treatment (7-year-old, watches other family member takes TB treatment)<br>Smells bad<br>Tablets are "*small*" | **Adverse consequences**<br>**- *Physiological***<br>Variety of minor adverse effects (nausea, stomach cramps, dizziness, insomnia, increased appetite)<br>Did not interfere with ability/willingness to administer drug<br>**- *Psychological***<br>Little associative stigma, some internalized stigma.<br>Caregivers use stigmatizing language towards index cases.<br><br>**Conceptions of health and illness**<br>Health requires regular water, fruit and veg and exercise<br>TB a contagious, airborne, "*terrible*" illness<br>MDR-TB "*a lot worse*", "*they say it sends you to your maker*"<br>Strong belief that TB preventive therapy will prevent TB<br><br>**Prior experiences of treatment and care**<br>Most families have substantial experience of TB – anxious to prevent in children - "*as long as my child is going to be alright*"<br>Little experience of preventive therapy in routine care | **Socioeconomic circumstances**<br><u>Barriers:</u> Most families struggling financially<br>Isolation, depression<br><u>Facilitators:</u><br>Free transport, financial compensation "*I can buy food*"; helpful study staff "*They really care*"<br><br>**Health system delivery**<br>**- *Accessing care on study***<br><u>Barriers:</u> Waiting times, blood draws, communication with drivers<br><u>Facilitators:</u> Accessible study sites, shorter waiting times (compared with routine care), sick certificates, convenient appointment times<br>**- *Accessing care in routine health system***<br><u>Barriers:</u> Loss of patient folders, long waiting times "*You sit there the whole day*", fear of contracting illness while waiting in queues, unavailability of certain medications, shortage of staff, unhelpful staff - "*they don't have passion*"<br><u>Facilitators:</u> None mentioned |

**Fig 1. Diagrammatic depiction of conceptual framework of TB treatment acceptability and key interview findings.**

TB, across three generations. Interviewees described TB as a contagious, "*terrible sickness*" that can affect people of all ages. Most caregivers and children understood that TB is an airborne disease. Five interviewees identified smoking as a risk factor for TB, while others associated TB transmission with cold weather, being indoors with closed windows and working on construction sites or in the mines. They were afraid of death from TB but one noted "*there's treatment you can take and be well, if you want*". MDR-TB was described as "*more dangerous*", or "*a lot worse*" than 'normal' TB. One caregiver reported, "*They say it sends you to your maker, if you don't take your medication*". Knowledge and prior experience of TB made caregivers anxious to do all they could to prevent TB in their children.

Interviewees felt they knew how to prevent TB, and suggested opening windows, not smoking, washing hands, avoiding busy places, wearing a mask, being careful who you are with and ensuring the index patient takes TB treatment correctly as ways to avoid being infected. Many of the interviewees admitted not knowing much about TPT in routine care, though a few either had family members who had taken TPT, knew people who received TPT from the clinic, or had taken TPT themselves. Participants understood and embraced the concept of TPT and believed it could keep them and their children well. Concerns about TPT generally included side-effects and "*Sometimes you have a lot of things on your mind, and you forget to take your prevention tablets*".

Although participants expressed not knowing a lot about TPT, most felt sure the study drug would protect their children from developing MDR-TB. One caregiver was very angry that her child needed the study drug, saying "*my child is going to take tablets because of an adult who deliberately did that to themselves*". This caregiver noted that the index patient was diagnosed after smoking cigarettes and cannabis with a friend who had MDR-TB. Intersectional stigma like this, where the index case is perceived as partaking in risky behaviour or as being irresponsible was common. A teenage girl (15-year-old) who became pregnant while on study also described being angry with the index patient, her uncle, "*because he doesn't want to take his treatment and he is putting our lives at risk*". She went on to say that she was afraid of losing her baby due to side effects of the study drug.

Receptivity to study drug was influenced by caregiver's fear of adverse consequences: "*Sometimes I wonder what are the side effects of the pills, but my daughter has to drink it for her health*" (caregiver to a 2-year-old). Caregivers and children noted a variety of minor adverse effects, but none of these interfered with their ability to administer or take the drug, and were present in both the levofloxacin and placebo arm. One participant said the neighbours noticed a change in her child's behaviour/personality after starting TPT, which made them curious as to what was happening in the household. The participant explained:

> "*It's that they see that he eats a lot, so they ask 'What's wrong with your child? Every 2 minutes he says he's hungry. Every 2 minutes he's hungry?' so I then explained that he is [on] TB preventive therapy, that's why he has this much appetite.*"

Most caregivers and children were happy to tell family, close friends, teachers and some neighbours about the study, and generally found people understanding and supportive. Caregivers did, however, stress to friends and family that their children did not have TB themselves, so that "*people understand it's prevention*". One 12-year-old said she did not tell her friends as she was afraid of judgement–"*They will say I don't want to play with you because you have sick family and I am going to get sick*". Fear of stigma/othering compromised some participants' willingness to administer treatment to children in the presence of their friends. A 17-year-old did not want her friends to think she was ill, so skipped taking her medication when friends were visiting. MDR-TB seemed more stigmatised than drug-susceptible TB, and it was clear

from the interviews that many of the index cases had not disclosed their MDR-TB status beyond their immediate household.

### Integration: Socioeconomic circumstances, health system delivery

Interviewees identified various barriers to accessing study drug while in the trial. Barriers included waiting times at the study site, regular blood draws and fear of needles and difficulties communicating with study drivers. Facilitators included free transport to the study site, reimbursement (*"makes you want to come" "I can buy food" "It's a big help for me"*), accessibility of study sites, shorter waiting times (compared with routine care), sick certificates given for school and work, being able to attend early and not miss school, not having to sit at clinics with large numbers of ill people, and supportive, hardworking and helpful staff (*"They really care"*). Study staff were contrasted with clinic staff who *"don't have passion"* and only attend to patients *"when they feel like it"*. Loss of physical patient folders (which occurs regularly in primary health care facilities in South Africa) caused long waiting times and considerable frustration for participants accessing TB services in routine care. It was clear that regular monthly attendance at busy routine health care facilities would place a significant burden on families, who would need to fund transport for these visits, and deal with the consequences of missing a full day of work, school or child-care obligations.

Researchers noted that many participants struggled to verbalise any hopes or dreams, and those expressed related mainly to attaining financial security. Some interviewees expressed disappointment as to how their lives had turned out–*"I had dreams but sometimes it doesn't work out"* (64-year-old caregiver) *"I am a low-class version of myself"* (22-year-old mother to 2-year-old participant). Researchers also noted substantial isolation in several participants—*"we are so rarely in the company of other people"; "I keep my distance for their safety"; "I don't do much, I am just bored at home"*. One caregiver seldom left the house and slept for more than 13 hours each day. Isolation, depression and other mental health issues may impact caregivers' ability to access and adhere to monthly TPT from routine care.

## Discussion

Ideally, young children should have access to medicines that are specifically designed for paediatric use [19]. There is increasing awareness that acceptability of medicines in children and adolescents is complex and related to multiple intrinsic and extrinsic factors [20]. Acceptability of a medication used in children should therefore be studied in children themselves. Standardised methods to assess the acceptability of medicines in children have not been established and there is currently little guidance in the literature on how to conduct or report acceptability testing in children [21, 22]. There are considerable gaps in current research, which has failed to account for the influence of caregivers' and children's contexts on treatment uptake and overall acceptability. We used the Wademan conceptual framework to ensure we explored the concept of the acceptability of adult 250mg levofloxacin tablets holistically.

We found that the adult levofloxacin tablet was generally acceptable in children who were able to swallow it whole. Younger children and their caregivers, however, complained about its taste when crushed or softened with a liquid. This is consistent with previous work which found relatively poor acceptability of the 250mg formulation when crushed or softened and better acceptability of a 100mg dispersible formulation [23, 24]. Some caregivers had to bribe or restrain their children, and felt the poor taste was a barrier to ongoing adherence. Some did note that the children became accustomed to the taste over time. Adverse effects were generally minor and were noted by participants in both the levofloxacin and the placebo arm. The use of

palatable 100mg dispersible paediatric levofloxacin formulations should thus be prioritized for TB care, particularly in young children unable to swallow the adult formulation.

Most caregivers and participants had previous experience of TB within their families and had seen the disease first-hand, which motivated their desire to protect themselves/their children. This is consistent with other research [25–27]. Participants believed strongly in the ability of the study drug to prevent them/their children from getting TB. Participants consistently listed financial compensation, free transport, and short waiting times as facilitators to accessing care. Fear of stigma, social isolation and poor socioeconomic circumstances were common in these families and should be considered when planning prevention strategies.

Strengths of our analysis are that the data were collected in-depth and longitudinally over the course of children's treatment experiences and included both caregivers and older children. The holistic acceptability framework proved a useful tool to guide interview preparation and data analysis, and its use ensured a broad evaluation of acceptability [17]. We purposefully sampled for diversity of age, gender, and home language, although men (as caregivers) were under-represented. Limitations include that the study was conducted at a single site, in a clinical trial setting, that the sample size was modest (we felt we reached saturation overall but not disaggregated by age or gender) and we were unable to sample by study arm.

Older children who swallowed the tablet whole found the adult levofloxacin formulation acceptable. However, facilitating access to a dispersible palatable levofloxacin formulation is key for very young children, who cannot swallow tablets and have the highest risk of disease progression. Access to better formulations, however, will not address the challenging home circumstances that many families face. Implementation models for MDR TPT must interface with the financial and social circumstances of the child, their caregiver and household context.

Acceptability work is frequently neglected in trial design and implementation, even though it is well known that acceptability of a drug and its formulations can directly impact adherence, intended use, efficacy and safety. It is important that evaluation of acceptability is planned early and undertaken holistically in children affected by TB and includes in-depth qualitative work alongside quantitative analysis. This is also increasingly viewed as important in the development of international guideline processes. More work on the acceptability of TPT and its related processes (contact tracing, household screening, and regular follow-up) should be conducted to inform policy and programme development.

## Supporting information

**S1 Text. Child-caregiver discussion guide.**
(DOCX)

## Acknowledgments

The authors thank the staff and study participants of the TB-CHAMP study, Cape Town, South Africa.

## Author Contributions

**Conceptualization:** Susan E. Purchase, Dillon T. Wademan, Graeme Hoddinott, James A. Seddon, H. Simon Schaaf, Anneke C. Hesseling.

**Data curation:** Susan E. Purchase, Dillon T. Wademan, Nosibusiso L. Tshetu, Mohhadiah Rafique.

**Formal analysis:** Susan E. Purchase, Dillon T. Wademan.

**Investigation:** Susan E. Purchase, Dillon T. Wademan, Nosibusiso L. Tshetu, Mohhadiah Rafique.

**Methodology:** Susan E. Purchase, Dillon T. Wademan, Graeme Hoddinott.

**Project administration:** Dillon T. Wademan, Nosibusiso L. Tshetu, Mohhadiah Rafique.

**Supervision:** James A. Seddon, H. Simon Schaaf, Anneke C. Hesseling.

**Writing – original draft:** Susan E. Purchase.

**Writing – review & editing:** Susan E. Purchase, Dillon T. Wademan, Nosibusiso L. Tshetu, Mohhadiah Rafique, Graeme Hoddinott, James A. Seddon, H. Simon Schaaf, Anneke C. Hesseling.

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
