## [Decision Letter · Decision Letter 0]

20 Feb 2024

PGPH-D-23-02619

Acceptability of an adult levofloxacin formulation in children and adolescents on a tuberculosis preventive treatment trial

Dear Dr. PURCHASE,

Thank you for submitting your manuscript to PLOS Global Public Health. After careful consideration, we feel that it has merit but does not fully meet PLOS Global Public Health’s publication criteria as it currently stands. Therefore, we invite you to submit a revised version of the manuscript that addresses the points raised during the review process.

Please address the reviewers' comments, especially the major concerns of reviewer 1.

We look forward to receiving your revised manuscript.

Kind regards,

Helen Jenkins

Academic Editor

Journal Requirements:

Additional Editor Comments (if provided):

Reviewers' comments:

Reviewer's Responses to Questions

**Comments to the Author**

1. Does this manuscript meet PLOS Global Public Health’s publication criteria? Is the manuscript technically sound, and do the data support the conclusions? The manuscript must describe methodologically and ethically rigorous research with conclusions that are appropriately drawn based on the data presented.

Reviewer #1: Yes

Reviewer #2: Yes

2. Has the statistical analysis been performed appropriately and rigorously?

Reviewer #1: N/A

Reviewer #2: N/A

3. Have the authors made all data underlying the findings in their manuscript fully available (please refer to the Data Availability Statement at the start of the manuscript PDF file)?

Reviewer #1: Yes

Reviewer #2: No

4. Is the manuscript presented in an intelligible fashion and written in standard English?

Reviewer #1: Yes

Reviewer #2: Yes

5. Review Comments to the Author

Reviewer #1: This is an excellent paper on an important topic, and the authors have done exceptional work including a qualitative acceptability study within a larger trial of TPT. The longitudinal nature of the interviews is a great strength, and it is also exciting to see that they included interviews with both younger and older children.

There are some areas of the paper, however, that need to be strengthened.

First, the authors cite the study by Wademan et al as the conceptual framework for the data analysis. This is a strong framework to use, but some readers may not be familiar with it. It would help (if there is space) to include the figure from this paper to guide the reader. It would also greatly strengthen the paper to have the discussion section organized around this framework and to include a figure of the findings as they related to each of the domains of acceptability identified in Wademan's paper. The Wademan framework is really seminal in the pediatric TB space. I know the framework well and understand the authors are presenting many of the acceptability aspects in the results/discussion section. But for readers not as familiar with the framework, the results seems a bit like a "laundry list" of issues. It is always a struggle with word limits and qualitative data, and a figure (with some examples/sample quotes) showacse the findings in the different acceptability domains would be a strong addition to the paper.

Second, I am not sure why the authors seem to overly focus on the dug formulation in the paper. I know this is an important issue, but it seems to me the authors did key work on the acceptability of TPT as a whole. And I think this is important to highlight in addition to the formulation work. TPT is often viewed as a secondary priority for programs, but for families affected by TB and DR-TB, prevention seems like a key priority from the data you present. I would highlight this more and not just focus on the formulation.

Third, while the focus of the paper (based on the title and framing) seems to be on the acceptability of the adult formulation, it is not clear to me why the authors feel it is necessary to see if adult formulations are acceptable to children. Data from multiple other diseases show that they clearly are not. So why do we need to prove this again in TB? I applaud the authors for doing this work and collecting the data, but I think they need to provide more framing and support for why we need data on the unacceptability of adult formulations for children. Some of this is about logistics and advocacy. But this should be acknowledged in the discussion--TB is way behind. There has been important work on pediatric formulation development in recent years in TB, however, and the authors should cite and acknowledge this more. The reader is also left to wonder why this study did not use the available pediatric formulations of levofloxacin (which were available for most of the study) or even try to compare the adult and the peds formulation. If this was for logistical reasons, this should be acknowledged more and discussed in the paper, as it seems like a hole in the study.

Some smaller comments are also below:

1. The authors should add a reference on the topic of saturation. This topic is somewhat controversial in qualitative research, and it should be referenced in the paper.

2. Is there a way to share the interview guides? I did not see them with the paper. Perhaps as an annex?

3. Lines 219-220 seem to be out of place here or missing some words?

4. Lines 232-233 refer to a "lack of caregiver ambition...". The authors should rethink this sentence. The previous sentences describe aspects of depression in people (sleeping for many hours, anhedonia). To see it called a "lack of ambition" seems a bit of a mis-judgement that could inadvertently contribute to stigma.

Overall, this is a strong paper that will add a great deal to field should the authors be able to make these revisions.

Reviewer #2: This is an important and well written paper about a sub-study of the TB-Champ trial by a very experienced and knowledgeable team both on clinical/clinical research aspects and acceptability studies in the field of childhood MDR TB treatment. The TB-Champ trial, together with the V-QUIN trial recently provided evidence that levofloxacin can prevent development of MDR-TB in household contacts of adults with MDR-TB (results presented at Union conference). Access to MDR TB drugs in children for TPT and MDR Tx is a key issue.

I only have a minor comments to consider.

1. Interviews and results are touching on many aspects, with a broad vision of acceptability, not restricted to palatability/acceptability of levofloxacin tablets per se, in line with Wademan’s conceptual framework. I suggest taking that in consideration in the title, which for me reads as focusing very much on the adult levofloxacin formulation hence on the 1st part of the results.

2. I suggest adding a short description of the instructions that were given to parents/children on how and when to take the pills as well as on the treatment education package received which may have influenced the perceptions that study participants had of their treatment. Results section includes a description on how caregivers administered the treatment but it would be helpful to understand what instructions were given to them.

3. Could authors clarify how they did purposive sampling/checked that they reached saturation with the case study longitudinal design? Was the 1st visit considered or did they follow children till the third interview to decide whether other recruitments were done (which would seem more challenging from a practical point of view).

4. Could authors clarify whether caregivers were the index case? This may also have influenced their perception of how the child dealt with treatment, although it appears from the interviews that within community stigma seemed limited.

5. Fear of stigma is mentioned once but there does not seem to be major stigma related to MDR TB as compared to DS TB, at least not in the way it is reported here. In the authors’ opinion, could this be due to the trial context and to a “selected” study population or is the community cluster randomized design representative of what is happening in the community?

6. At last, in the discussion, authors refer to the better acceptability of levofloxacine 100 mg. Was this considered as an alternative in the trial? If no why not?

6. PLOS authors have the option to publish the peer review history of their article (what does this mean?). If published, this will include your full peer review and any attached files.

**Do you want your identity to be public for this peer review?** For information about this choice, including consent withdrawal, please see our Privacy Policy.

Reviewer #1: No

Reviewer #2: **Yes: **Olivier Marcy

---

## [Decision Letter · Decision Letter 1]

31 May 2024

Holistic acceptability of an adult levofloxacin formulation in children and adolescents on a tuberculosis preventive treatment trial

PGPH-D-23-02619R1

Dear Dr PURCHASE,

We are pleased to inform you that your manuscript 'Holistic acceptability of an adult levofloxacin formulation in children and adolescents on a tuberculosis preventive treatment trial' has been provisionally accepted for publication in PLOS Global Public Health.

Best regards,

Helen Jenkins

Academic Editor

Reviewer Comments (if any, and for reference):

Reviewer's Responses to Questions

**Comments to the Author**

1. If the authors have adequately addressed your comments raised in a previous round of review and you feel that this manuscript is now acceptable for publication, you may indicate that here to bypass the “Comments to the Author” section, enter your conflict of interest statement in the “Confidential to Editor” section, and submit your "Accept" recommendation.

Reviewer #1: All comments have been addressed

2. Does this manuscript meet PLOS Global Public Health’s publication criteria? Is the manuscript technically sound, and do the data support the conclusions? The manuscript must describe methodologically and ethically rigorous research with conclusions that are appropriately drawn based on the data presented.

Reviewer #1: Yes

3. Has the statistical analysis been performed appropriately and rigorously?

Reviewer #1: N/A

4. Have the authors made all data underlying the findings in their manuscript fully available (please refer to the Data Availability Statement at the start of the manuscript PDF file)?

Reviewer #1: Yes

5. Is the manuscript presented in an intelligible fashion and written in standard English?

Reviewer #1: Yes

6. Review Comments to the Author

Reviewer #1: Thanks for addressing all the concerns and for doing this outstanding work.

7. PLOS authors have the option to publish the peer review history of their article (what does this mean?). If published, this will include your full peer review and any attached files.

**Do you want your identity to be public for this peer review?** For information about this choice, including consent withdrawal, please see our Privacy Policy.

Reviewer #1: No
